# Analyzing the Spatiotemporal Uncertainty in Urbanization Predictions

Jairo Alejandro Gómez [1,†], ChengHe Guan [2,*,†], Pratyush Tripathy [3,†], Juan Carlos Duque [4,†], Santiago Passos [4,†], Michael Keith [5,†] and Jialin Liu [6,†]

1    i2t Research Group, Department of Communication and Information Technologies, Universidad Icesi, Calle 18 No. 122-135, Pance, Cali 760031, Colombia; jagomez3@icesi.edu.co
2    Arts and Sciences, New York University Shanghai, 1555 Century Avenue, Pudong New District, Shanghai 200122, China
3    Geospatial Lab, Indian Institute for Human Settlements, IIHS Bengaluru City Campus, 197/36, 2nd Main Road, Sadashivanagar, Bengaluru 560080, India; pratyush@iihs.ac.in
4    Research in Spatial Economics (RiSE) Group, Department of Mathematical Sciences, EAFIT University, Carrera 48 A 10 Sur 107, Casa 4, oficina RiSE, Medellín 050022, Colombia; jduquec1@eafit.edu.co (J.C.D.); spassos@eafit.edu.co (S.P.)
5    COMPAS, School of Anthropology, University of Oxford, 58 Banbury Road, Oxford OX2 6QS, UK; michael.keith@compas.ox.ac.uk
6    Key Laboratory of National Forestry and Grassland Administration on Ecological Landscaping of Challenging Urban Sites, Shanghai Engineering Research Center of Landscaping on Challenging Urban Sites, Shanghai Academy of Landscape Architecture Science and Planning, 899 Longwu Road, Xuhui District, Shanghai 200232, China; jialin.liu@nyu.edu
*    Correspondence: chenghe.guan@nyu.edu; Tel.: +86-21-2059-5500
†    These authors contributed equally to this work.

**Abstract:** With the availability of computational resources, geographical information systems, and remote sensing data, urban growth modeling has become a viable tool for predicting urbanization of cities and towns, regions, and nations around the world. This information allows policy makers, urban planners, environmental and civil organizations to make investments, design infrastructure, extend public utility networks, plan housing solutions, and mitigate adverse environmental impacts. Despite its importance, urban growth models often discard the spatiotemporal uncertainties in their prediction estimates. In this paper, we analyzed the uncertainty in the urban land predictions by comparing the outcomes of two different growth models, one based on a widely applied cellular automata model known as the SLEUTH CA and the other one based on a previously published machine learning framework. We selected these two models because they are complementary, the first is based on human knowledge and pre-defined and understandable policies while the second is more data-driven and might be less influenced by any a priori knowledge or bias. To test our methodology, we chose the cities of Jiaxing and Lishui in China because they are representative of new town planning policies and have different characteristics in terms of land extension, geographical conditions, growth rates, and economic drivers. We focused on the spatiotemporal uncertainty, understood as the inherent doubt in the predictions of where and when will a piece of land become urban, using the concepts of certainty area in space and certainty area in time. The proposed analyses in this paper aim to contribute to better urban planning exercises, and they can be extended to other cities worldwide.

**Keywords:** urban growth; urbanization processes; spatiotemporal uncertainty; urban planning tools; urban science; SLEUTH CA; machine learning

## 1. Introduction

Recent statistics from the United Nations show that humans are now urban species with more people living in cities than in rural areas [1,2]. In China, rapid urbanization

has completely reshaped cities and towns. After more than thirty years of continuous growth, China has entered into a relatively stable period of urbanization [3,4]. Now, smaller cities and towns -those with fewer than 5 million people- are gaining more attention and attracting new opportunities as regional planning policies are redistributing resources away from larger cities [5]. Today's planning policy in China aims for a balanced network of cities, where the small cities and towns in the coastal region become key components of regional urban networks, releasing the pressure, congestion, and pollution of larger cities such as Shanghai, Beijing, and Guangzhou. Overall, small cities face greater growth uncertainties than large cities, making it harder for city planners and other stakeholders to anticipate their future patterns of urbanization. Here, by urban growth uncertainty, we refer to the difficulty in predicting the exact type, location, occurrence time, and development stages of future urbanization [6]. However, in this paper, we focus exclusively on the spatiotemporal urban growth uncertainty, which is associated with where and when a piece of land becomes urban. The spatial uncertainty in urban growth predictions under different scenarios increases as there is more available land for urban expansion. Similarly, the temporal uncertainty increases when we predict further ahead in the future because of the inevitable integration process of noise in the input variables, and changes that always happen in politics, economics, environment, and social contexts, just to name a few. Despite the challenges, we argue that by including spatiotemporal uncertainties in urban growth predictions, stakeholders such as policy makers and urban planners can be better equipped to comprehend the urbanization process, recommend a better course of action, and help policy makers to prioritize public interventions. The spatiotemporal uncertainty analysis of urban growth has been addressed in the forecast of population [7], magnitude of urban sprawl and expansion, economic growth [8], and the impact on environment sustainability [6]. In this paper, we analyze the spatiotemporal uncertainty of future urbanization processes by exploring the agreements and disagreements of two urban growth models, which have distinct variable selection processes, growth algorithms, simulation mechanisms, and performance indicators. In this regard, the literature from different disciplines that focuses on growth models is extensive [9–18], it is worth mentioning the urban-oriented growth models have been applied to predict urban futures since the 1970s [19]. In most models, proper calibration and training using historical data are necessary to capture the trajectory and trend of urban growth [20]. Early urban growth models suffered from insufficient computational power and applied brute-force algorithms. The emergence of new modelling schemes, higher resolution data, and better computing capabilities enabled the progress of urban growth modelling. Recognizing that urban growth is a dynamic process with high uncertainty, [21] adopted the outcomes of planning policy to the model, [5] incorporated growth-constrain rules into the urban growth models, and [22] created multiple scenarios to simulate zone-specific land-use plans. To perform the spatiotemporal uncertainty analysis in this paper, we use a cellular automata (CA) model known as SLEUTH CA [20], and a Machine Learning (ML) framework that was introduced recently in [23]. The SLEUTH CA model is discrete in time, space, and state [24] and has been successfully applied worldwide to simulate land-use change [25]. The SLEUTH model considers slope of terrains, land use types, excluded areas for urban growth, urbanized areas, and transportation road networks. Data entry is in the form of a pixelated mapped depiction, where pixels again correspond to cells in the Cellular Automata scheme. Generally, the calibration process included a coarse calibration, a fine calibration, and a final calibration. In turn, the ML framework models the urban growth and provides predictions of three variables corresponding to the spatially distributed population, binary urban footprint (i.e.,: spatial distribution of urban and non-urban areas), and urban footprint in color (i.e.,: the visual appearance of the territory in RGB color). The ML framework models the population distribution as a spatiotemporal dynamic system using a multiple-input single-output regression, furthermore, it obtains the binary urban footprint from the population distribution through a binary classifier and then adds a temporal correction for existing urban regions. In the last step, the ML framework esti-

mates the urban footprint in color from its previous value, as well as from past and current values of the binary urban footprint using a semantic inpainting algorithm. We selected the SLEUTH CA model and the ML framework because they are complimentary, the first is based on human knowledge and pre-defined and understandable policies while the second is purely data-driven and might be less influenced by any a priori bias. Table 1 provides a comparison of the main characteristics of these two models.

**Table 1.** Comparison between the SLEUTH CA model and the ML-based urban growth framework. SAD stands for sum of absolute differences. SSD stands for sum of squared differences. ZNCC stands for zero-mean normalized cross-correlation. FP rate stands for false-positive rate. IoU stands for intersection over union.

| Characteristics | SLEUTH CA | ML Framework |
|---|---|---|
| Does it expect the input variables in raster format? | Yes | Yes |
| Does it encode human intuition or human rules? | Yes | No |
| Does it require manual tuning or calibration? | Yes | No |
| Is it model driven or data driven? | Model (and data) driven | Data driven |
| Can it include additional independent variables with ease? | Yes with additional rules | Yes |
| What does the output depend on? | Output depends only on previous state(s) | Output depends on previous state(s) and inputs |
| Does it support future deterministic urban interventions? | Not in the standard implementation | Yes |
| Does it use population distribution as the main urban-growth driver? | No. Require manual input | Yes |
| Does the prediction change every time the model is run? | Yes | No |
| What is the key performance indicator of model fitness? | Shape index that measures the spatial fit between the model's growth | For population distribution: SAD, SSD, RMSE, Pearson's correlation coefficient. For binary urban footprint: FP rate, error rate, ZNCC, accuracy, $F_1$ score, and the IoU, which is also known as the Jaccard index |
| Does it require expert knowledge to calibrate or train the model? | Yes | No |
| Does the model estimate get better with more training data? | Yes or no. Depending on the time and space dimension of data | Yes |
| Is it widely used in the literature? | Yes | Not yet, it is very recent |

To test our methodology, we pick two small cities in China with divergent geographical conditions, known as Jiaxing and Lishui. These cities are located in Zhejiang Province, where the policy on "the in-depth stage of transformation and development of small cities (TDSC)" announced during the eleventh five-year plan (2006–2010) gravitated towards small cities and towns with strong economic infrastructure [26,27]. Jiaxing is located in an alluvial plain and Lishui in a riverine valley. The alluvial plain at the estuary of the Qiantang River provides Jiaxing relatively less constrain for urban growth while the riverine valley of Daxi Brook leaves very limited land for Lishui's future urbanization. According to the 2017 household registration [28] data, Jiaxing with 3.56 million people and Lishui with 2.69 million people ranked 156 and 202 out of the over 300 prefecture-level cities in the country in terms of population size, respectively. As mentioned earlier, small cities like Jiaxing and Lishui forfeited their growth opportunity to large cities in the early stage of China's rapid urbanization. Nowadays, they are gaining momentum to rapid urban growth represented by land expansion, increasing migrant population, and economic growth. To analyze the urbanization uncertainty in these cities, we compute

annual predictions from 2016 to 2040 and quantify the spatial and temporal disagreement of the binary urban footprint estimates.

The rest of this paper is organized as follows. Section 2 describes the materials and methods for acquiring and pre-processing the data for Jiaxing and Lishui, as well as the procedure for calibrating the SLEUTH CA model and for training and validating the ML framework. Section 3 presents the results of the urban growth predictions in these two cities as well as some key graphs and metrics to quantify their spatial and temporal differences and hence the uncertainty. Section 4 includes a brief discussion of the obtained results in the context of China and more general in terms of urban planning. Finally, Section 5 presents the conclusions of this paper, its limitations, and future work.

## 2. Materials and Methods

In Section 2.1 we describe the input data for the SLEUTH CA model and the ML framework. As each approach requires different variables and assumptions, we go through the pre-processing and calibration of the SLEUTH CA model in Section 2.2, and the data processing, training, validation, and testing of the ML urban growth framework in Section 2.3. Finally, we introduce the proposed analyses to quantify the spatial and temporal uncertainty based on the urban-growth prediction disagreements in Section 2.4.

### 2.1. Input Data for Urban Growth Modeling

To perform the spatial and temporal uncertainty analyses of urban growth predictions, we selected two small cities, Jiaxing and Lishui. In both cases, we focused on the central urban districts of the prefecture-level cities. Jiaxing is a small city located in northern Zhejiang Province and its built-up area holds two urban districts with 1,201,800 inhabitants [29]. Jiaxing is known as the hometown of silk, it has a strong industrial base of textiles, and a state-level export processing zone for IT, electronics, and other high-tech industries. Importantly, Jiaxing is on the route of a busy high-speed Shanghai-Hangzhou passenger railway corridor. In contrast, Lishui is located in southern Zhejiang Province with one urban district of 351,000 inhabitants. In general, both cities have very different geographical conditions as Jiaxing is close to large cities such as Hangzhou, whereas Lishui is far away from the large urban centers and from the coastal line. Figure 1 shows the locations of Jiaxing and Lishui in the Zhejiang Province in China, as well as their visual appearance captured from satellite imagery.

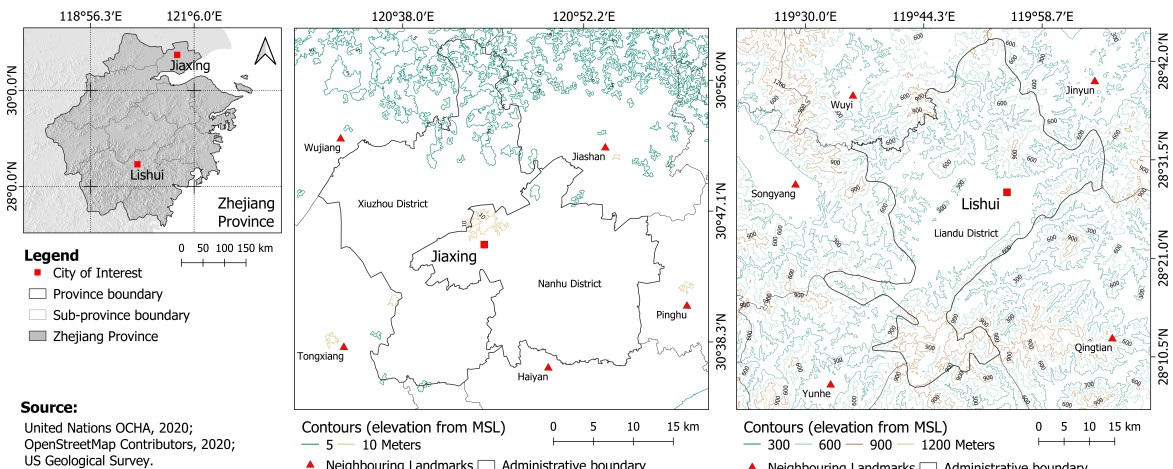

**Figure 1.** Location and images of the cities of Jiaxing and Lishui in the Zhejiang Province in China.

After selecting the two cities, the next step was to extract the available historical variables to feed the two urban growth models. Table 2 summarizes this process, describing the name of each variable, its source, digital format, original spatial resolution, temporal availability, and the model that uses it. In this paper, both models use the binary urban

footprint, land use, terrain slope, and water bodies. The latter are used as exclusion areas for urbanization. Only the SLEUTH model uses the hillshade, and road map. In turn, the ML framework uses the remaining variables, including population distribution, maximum population capacity, official population projections, and potential of roads. In this paper, the potential $V$ of an arbitrary variable $v_i$ is computed through Equation (1). In this equation, $x$ and $y$ are spatial coordinate indices, $t$ is a discrete variable representing the time in years, and $x_p$ and $y_p$ represent the spatial indices where the potential is computed. Equation (1) was inspired by the electric potential used in Physics, but it was adapted to the urban context to avoid singularities that could occur otherwise when $x = x_p$ and $y = y_p$. Similarly to its counterpart in Physics, our definition adds up the potential contributions from all pixels in the space, providing large "potential values" for points in space located close to others where the input variable has positive values. Notice that in our convention, road pixels have values of one, while non-road pixels have values of zero.

$$V\left[v_i\left(x_p, y_p, t\right)\right] = \sum_{\forall x \neq x_p} \sum_{\forall y \neq y_p} \left( \frac{v_i(x, y, t)}{\sqrt{\left(x - x_p\right)^2 + \left(y - y_p\right)^2}} \right) \tag{1}$$

As it was impossible to get all the required input variables in Table 2 over a common period, we collected them for various years between 1990 and 2019 based on their temporal availability. Then, as a first pre-processing stage, we converted all the variables to rasters, ensuring that they shared the same coordinate reference system and geographic extent for each city, as well as a spatial resolution of $100\,\text{m} \times 100\,\text{m}$. We had to select a shared spatial resolution because both urban growth models require their input variables defined over the same spatial lattice. By looking at the variables, we saw that some of them like Landsat images had a $30\,\text{m} \times 30\,\text{m}$ resolution, while others like the spatial population distribution could have up to $250\,\text{m} \times 250\,\text{m}$. So, we decided to use an intermediate $100\,\text{m} \times 100\,\text{m}$ resolution, which is very convenient as it allows us to explain the "pixel" to policymakers as representing a hectare of land. To get this intermediate and shared spatial resolution, we applied a content-aware spatial resampling on the variables. However, this process does not improve the worst spatial resolution nor its accuracy. It is worth highlighting that the spatial resampling process affects the ML framework more than the SLEUTH CA as the latter does not use the coarse spatial-population-distribution variable. After this stage, Jiaxing's variables had 451 rows × 442 columns, and Lishui's rasters had 704 rows × 518 columns. The resulting rasters were stored as .gif images for the SLEUTH CA model and geo-referenced .tif rasters (GeoTIFF) for the ML framework.

**Table 2.** Variables that we used for the modeling the urban-growth. GHSL = Global Human Settlement Layer; LULCC = Land-Uses and Land-Cover Changes; IGSNRR = China's Institute of Geographic Sciences and Natural Resources Research; SRTM = Shuttle Radar Topography Mission; DEM = Digital Elevation Map; OSM = Open Street Map.

| Variable Name | Data Source | Digital Format | Original Resolution | Temporal Availability | Used by SLEUTH CA | Used by ML Framework |
|---|---|---|---|---|---|---|
| Binary urban footprint | Obtained from [30] | Raster | 30 m × 30 m | 1990–2015 | Yes | Yes |
| Land use | Obtained from IGSNRR [31] through LULCC classifications | Raster | 100 m × 100 m | 1990, 2000, 2010 | Yes (Recoded using Anderson Level I classification—see Appendix B) | Yes (residential and industrial uses) |
| Terrain slope | Derived from DEM in SRTM [32] | Raster | 90 m × 90 m | 2015 | Yes | Yes |
| Hillshade | Derived from DEM in SRTM [32] | Raster | 90 m × 90 m | 2015 | Yes | No |
| Water bodies | Obtained from IGSNRR [31] through LULCC classifications | Raster | 100 m × 100 m | 1990, 2000, 2010 | Yes | Yes |
| Roads | OSM [33] | Vector | - | 2018–2019 | Yes | No |
| Potential of Roads | Derived from OSM [33] through Equation (1) | Raster | 100 m × 100 m | 2018–2019 | No | Yes |
| Population distribution | GHSL pop [34] | Raster | 250 m × 250 m | 1990, 2000, 2015 | No | Yes |
| Maximum population capacity | Derived from population distribution and binary urban footprint | Raster | 100 m × 100 m | 2015 | No | Yes |
| Official population projections | China Statistical Yearbooks [28,29] | Tabular | Administrative units | 1978–2018 for Jiaxing 1991–2018 for Lishui | No | Yes |

## 2.2. Sleuth CA Urban Growth Model

To perform the urban growth modelling using the SLEUTH CA model, we preprocessed the following layers mentioned in Table 2: slope, exclusion, urban, roads and hillshade. We extracted the binary water class for the years 1990, 2000, and 2010 from the land-use layers by merging four sub-classes viz.; lake, reservoir/pit, river and canal. We added the binary water layers and normalized the values of the resulting layer between 0 and 100. This new layer was considered as the exclusion layer, where pixels with a value of 100 represent impossible growth and lower values suggest undesirable growth. We converted the pixels with a value of 100 in the exclusion layer, to zero in the hillshade layer, representing perennial water body. We also converted all the GeoTIFF layers that had a resolution of 100 m × 100 m to images in GIF format using the software QGIS 3.4.12 LTR.

The SLEUTH model controls the growth using five factors viz.; diffusion, spread, breed, slope, and road gravity. The objective of the SLEUTH model is to obtain a set of coefficients that best replicate the actual growth given the current input data. This is achieved in three calibration steps and the additional coefficient derivation step. We initiated the coarse calibration with the default coefficient values (i.e.,: 0–100 with a step of 25) for all the five factors. At each calibration stage, we evaluated the performance of the coefficient values using the Optimum SLEUTH Metric (OSM) to sort the 'control_stats.log' file produced by the SLEUTH model, and to narrow down the range of the coefficients at each calibration step [35]. The OSM is a product of metrics known as compare, population, edges, clusters, slope, X-mean, and Y-mean. For succeeding calibration stages, we considered the top three rows sorted in descending order by the OSM product. We used different coefficient ranges and Monte Carlo (MC) iterations for both Jiaxing and Lishui at all calibration stages, please refer to Appendix A: Table A1 for more details about this process.

To derive the coefficients after the final calibration, we used the coefficient values from the top row of the table sorted in descending order by the OSM product. The SLEUTH CA software has the ability to self-adjust and modify the coefficients, and therefore, as a final step, we used the coefficient values for the year 2015 included in the 'avg.log' file for future urban growth predictions.

## 2.3. ML-Based Urban Growth Framework

The input variables that we used for modeling the urban growth in Jiaxing and Lishui with the ML framework were: population distribution, binary urban footprint, water bodies (used as protected areas), terrain slope, potential of roads, residential land, industrial land, and maximum population capacity. We started from a dataset that had information only for a few years. We processed the dataset following the steps that are described in [23] with a few exceptions. For instance, time series with the total official population projections were extended until 2040 using a logistic regression; the yearly binary urban footprints were obtained directly from [30] for the period between 1990 (despite it was available since 1985) and 2015, rather than getting them by applying a binary (urban vs. non-urban) classifier to Landsat images; and finally, as the maximum population capacity was not available, we approximated it with the spatially distributed population of 2015, and then we adjusted it to take into account the protected areas and to leave some capacity for the population to grow. In this regard, we computed the per-pixel maximum in the protected areas with a value of zero to reflect the fact that some people might actually live in some protected areas (particularly in developing countries). Then we took the previous intermediate result and in the non-protected areas, computed the per-pixel maximum with the average urban population of the same year, a process that ensures some capacity for the population to grow. After this pre-processing stage, we assembled a complete dataset that had yearly estimates for all the variables between 1990 and 2015. Afterwards, we split the dataset, and used the data from years 1990 to 2005 for training, 2006 to 2010 for model selection, and 2011 to 2015 for testing, which roughly corresponds to a 60%, 20%, 20% data split, respectively.

For the spatiotemporal regression of the population distribution, we used a spatial neighborhood of $p \times q = 3 \times 3$ pixels (i.e.,: 300 m×300 m) and explored two consecutive

temporal lags corresponding to $\phi = \{1, 2\}$ years for all variables. We configured an autotuning program [23] to explore linear regressions, Bayesian regressions, and Ridge regressions. For the latter models, the autotuning searched through different values for the regularization parameter, namely $\lambda = \{0.1, 0.3, 0.5, 0.7, 0.9\}$. The autotuning was set to use five iterations for each city. After training and validating the model, we computed the performance on the test set. As we expected to see the most notable differences between the urban predictions of the two models after two or three decades, we computed yearly urban growth predictions from 2016 to 2040.

*2.4. Uncertainty Analysis of Urban Growth*

After calibrating the SLEUTH CA model and training the ML framework, we created a few graphs that simplify the analysis of the spatiotemporal uncertainty in the urban growth predictions for Jiaxing and Lishui. These graphs include:

1.  The yearly urbanization from 2015 to 2040, where 2015 is the last historical year, and the data from 2016 to 2040 corresponds to predictions.
2.  An instantaneous spatial difference between the models for each decade, i.e.,: 2020, 2030, and 2040.
3.  The cumulative spatial agreement and disagreement of predicted urban areas from 2016 to 2040 removing the common urban areas that already existed in 2015 to ease the visual inspection.
4.  A histogram with the "signed" differences in predicted urbanization times between the models to reveal if one of the models tends to predict urbanization earlier than the other.
5.  A histogram with the "unsigned" differences in predicted urbanization times between the models to understand how long does it take them to reach an agreement within the simulation horizon.
6.  A time series showing the evolution of the spatial agreement of the urban predictions of both models over time excluding the existing urban areas of 2015. In this case, we propose an agreement index $S$ through Equations (2) and (3), where the sum occurs over all pixels, $t_0$ is the reference year (which in our case is equal to 2015 because it is the last historical year), the sub-index $i = 1$ refers to the predictions from the SLEUTH CA model, while $i = 2$ corresponds to the predictions from the ML urban growth framework, and *BUF* is an acronym for the binary urban footprint. In our convention, a value of one in BUF indicates an urban area, whereas a value of zero indicates a non-urban area. The proposed agreement index can be seen as a special variant of the intersection over union (IoU) that blocks de-urbanization effects and prevents over estimating the spatial agreement, as it is relative to the last known and common urbanization state for both models. Notice that when de-urbanization is prohibited from the forecasts of both models, Equation (3) reduces to Equation (4). Also notice that $S$ varies from zero to one as the two predicted binary urban footprints move away from a total spatial disagreement to a perfect spatial agreement.

$$S(t) = \frac{\sum_x \sum_y [I_1(x, y, t) \cap I_2(x, y, t)]}{\sum_x \sum_y [I_1(x, y, t) \cup I_2(x, y, t)]} \tag{2}$$

$$I_i(x, y, t) = \left( \bigcup_{j=t_0}^{t} BUF_i(x, y, j) \right) - BUF(x, y, t_0); \text{ if there can be de-urbanization.} \tag{3}$$

$$I_i(x, y, t) = BUF_i(x, y, t) - BUF(x, y, t_0); \text{ if there is no de-urbanization.} \tag{4}$$

7.  A time series of two frequently used indices in the literature. The first index is known as the zero-mean normalized cross-correlation (ZNCC), see Equations (5) and (6), where $E_{x,y}[.]$ is the expected-value operator computed as the arithmetic mean across all the spatial coordinates. The ZNCC can change from $-1$ to 1, and the closer it gets to 1, the better the spatial agreement.

$$ZNCC(t) = \frac{\sum_x \sum_y \left\{ \left[ BUF_1(x,y,t) - \overline{BUF_1}(t) \right] \left[ BUF_2(x,y,t) - \overline{BUF_2}(t) \right] \right\}}{\sqrt{\left\{ \sum_x \sum_y \left[ BUF_1(x,y,t) - \overline{BUF_1}(t) \right]^2 \right\} \left\{ \sum_x \sum_y \left[ BUF_2(x,y,t) - \overline{BUF_2}(t) \right]^2 \right\}}} \tag{5}$$

$$\overline{BUF_i}(t) = E_{x,y}[BUF_i(x,y,t)] \tag{6}$$

The second index is known as the sum of absolute differences (SAD), and we computed it using Equation (7). The SAD is greater than or equal to zero, and the closer to zero the better the spatial agreement.

$$SAD(t) = \sum_x \sum_y |BUF_1(x,y,t) - BUF_2(x,y,t)| \tag{7}$$

Notice that both indices assess the "instantaneous" level of spatial agreement between the urbanization predictions of both models over time, as they are computed directly over the yearly predictions, i.e.,: without discarding de-urbanization effects nor the urban areas that existed in 2015.

## 3. Results

### 3.1. Sleuth CA

We implemented the SLEUTH CA model in Cygwin 64 terminal on Intel Xeon(R) E-2246G (3.60 GHz and 6 cores), 64 GB RAM, Windows 10 Pro OS machine. The coefficients that we obtained for predicting urban extent for Jiaxing and Lishui are summarized in Table 3. These coefficients can vary from 0 to 100, where a higher number signifies greater influence. For example, the Slope coefficient for Lishui, which is situated in a valley, is significantly larger than Jiaxing, which is situated on a flat terrain. This shows that the Slope variable had a greater influence in case of Lishui, as compared to Jiaxing. However, this influence does not follow a linear trend, for instance, an increase of the slope coefficient from 1 to 68 could dramatically change the urban growth rate but a decrease from 27 to 17 may only have a limited impact on the outcome.

We found that the convergence of coefficient values, and hence the model training, was slower in case of Lishui. This can be observed in the top three rows of the `control_stats.log` file sorted in descending order using the OSM metric. For example the diffusion, slope & road gravity coefficient ranges were the same after coarse and fine calibration stages, which should ideally have narrowed after each calibration stage, see Appendix A: Table A2. In addition, we found that the OSM metric values were higher in Jiaxing than in Lishui, please refer to Appendix A: Table A1 for details of the coefficient ranges used at each step of calibration, coefficient derivation, and prediction. According to the predictions from the SLEUTH CA model, the urban area will grow from from 862.97 km$^2$ in 2020 to 1552.73 km$^2$ in 2040 for Jiaxing, and from 128.06 km$^2$ in 2020 to 173.40 km$^2$ in 2040 for Lishui.

**Table 3.** Coefficients of the SLEUTH CA models for Jiaxing and Lishui.

| Prediction Coefficient | Value for Jiaxing | Value for Lishui |
|---|---|---|
| Diffusion | 100 | 32 |
| Breed | 15 | 67 |
| Spread | 27 | 17 |
| Slope | 1 | 68 |
| Road Gravity | 74 | 96 |

### *3.2. ML-Based Urban Growth Framework*

We implemented the ML framework in Python 3.x in Ubuntu 18.04.4 LTS using a laptop with an Intel(R) Core(TM) i5-9300H CPU @2.40 GHz that had 4 cores and 8 sub-processes, 8 GB of RAM, and 64 GB of SWAP memory on hard disk. The overall training, model selection, testing, and simulation (i.e.,: computing the predictions from 2016 to 2040) of the ML framework for both cities took about 7 h. The detailed processing times for each city and stage are summarized in Table 4.

**Table 4.** Time spent by the ML framework to processing the cities of Jiaxing and Lishui.

| Task | Time for Jiaxing | Time for Lishui |
|---|---|---|
| Pre-processing | 6.31 min | 11.74 min |
| Training | 98.29 min | 176.15 min |
| Model selection | 6.60 min | 7.20 min |
| Test | 4.77 min | 5.77 min |
| Simulation | 58.79 min | 62.46 min |
| Total | 174.76 min | 263.32 min |

As part of the processing, the autotuning program found that the best models had a consecutive temporal lag $\phi = 2$ years. Likewise, it found that the best model for predicting the population distribution of Jiaxing was a Bayesian Ridge regression with default parameters, while for Lishui the best model was a Ridge regression with a regularization parameter of $\lambda = 0.9$ and a tolerance of $tol = 10^{-3}$. In terms of losses relative to the ground truth, the average sum of absolute differences (SAD) for the binary urban footprint estimations in the datasets of training, model selection, and testing were 48,174; 40,753.33; and 38,635.66 pixels for Jiaxing, and 7082.71; 6801.33; and 7516.66 pixels for Lishui, respectively. All the additional metrics that we computed for the ML-based urban growth framework for both cities are available in the Appendix B. According to the predictions from the ML framework for Jiaxing, the urban area will grow from 1043.84 km$^2$ in 2020 to 1078.13 km$^2$ in 2040. Likewise for Lishui, the urban area will grow very little from 155.33 km$^2$ to 155.39 km$^2$ in the same period.

### *3.3. Uncertainty Analysis of Urban Growth*

To understand the yearly predicted urbanization process of Jiaxing and Lishui according to the SLEUTH CA and ML growth models from 2015 to 2040, the reader is referred to Figures 2 and 3. These figures have been color-coded. Black areas correspond to non-urban land and orange areas, independent of the saturation, represent urban land. The saturation in the orange color of these figures reflects the predicted year of urbanization, i.e.,: the lighter the color the later the urbanization occurs.

Figures 4 and 5 show the historical binary urban footprint in 2015 and the predictions from the SLEUTH CA model (identified as M1) and the ML framework (identified as M2) every ten years between 2020 and 2040 for Jiaxing and Lishui, respectively. In these figures, for the first two columns we use a white color to represent urban land and a black color to represent non-urban land. For the third column, we use the white and black colors to show the corresponding urban and non-urban agreements, respectively. To highlight the spatial disagreements between the urban growth predictions for a given year in the third column, we use red and blue colors. The red color is used when the SLEUTH CA model

predicts non-urban land but the ML model predicts urban land, and the blue color is used when the SLEUTH CA model predicts urban land but the ML model predicts non-urban land instead.

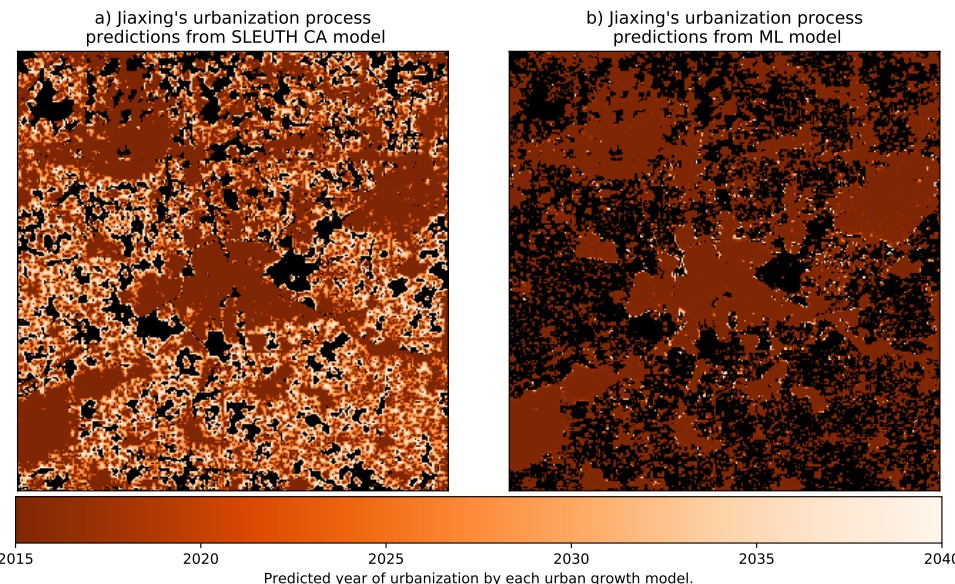

Note: Black pixels correspond to land areas where the model did not predict urbanization.

**Figure 2.** Yearly predicted urbanization by the SLEUTH CA and ML models for Jiaxing.

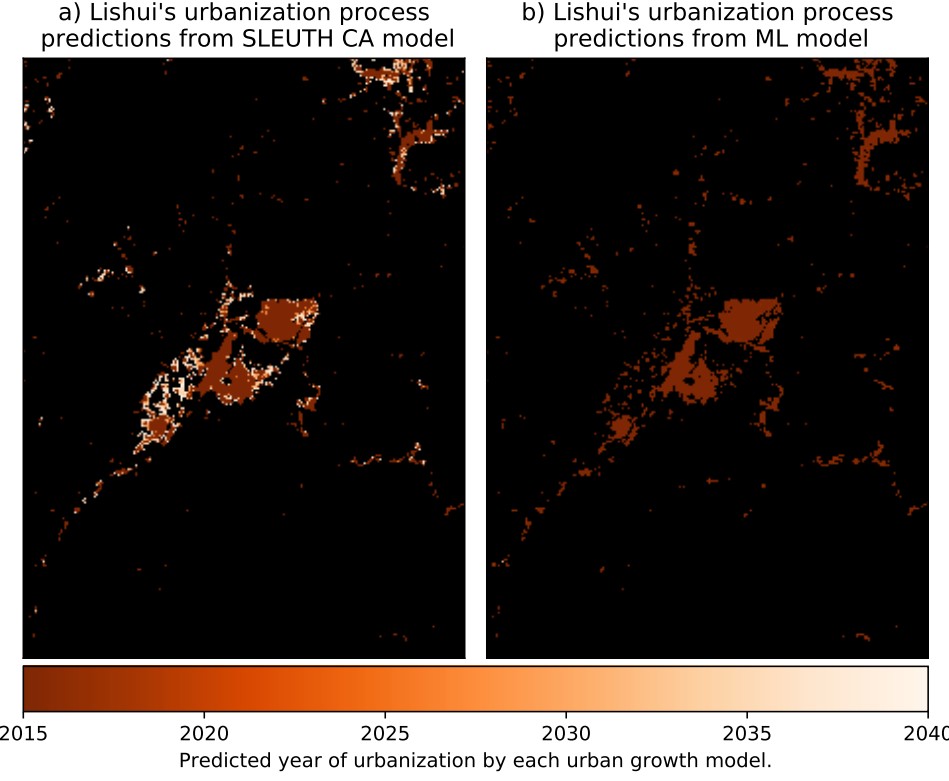

Note: Black pixels correspond to land areas where the model did not predict urbanization.

**Figure 3.** Yearly predicted urbanization by the SLEUTH CA and ML models for Lishui.

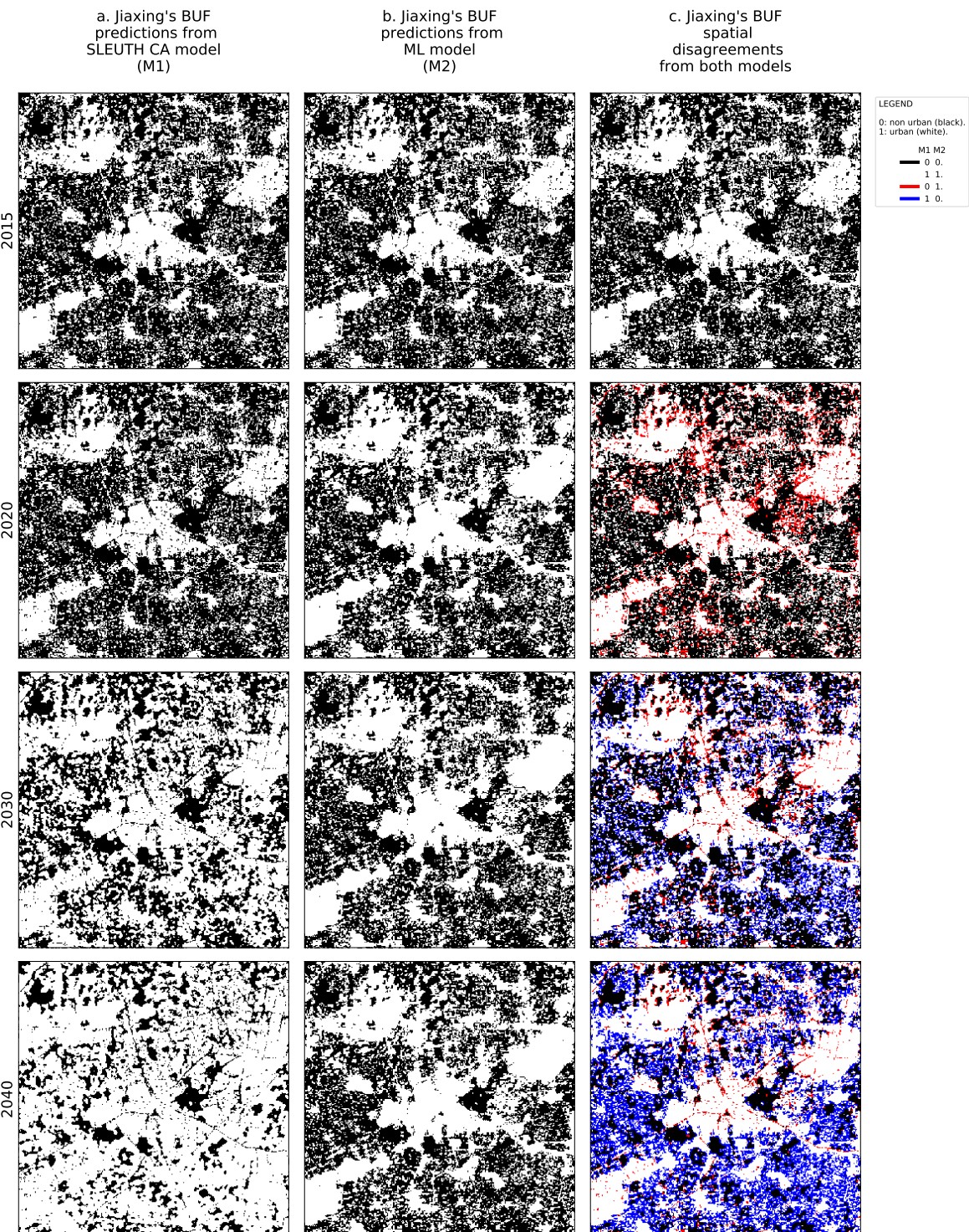

**Figure 4.** Yearly binary urban footprint predictions and spatial disagreements between the SLEUTH CA and ML models for Jiaxing.

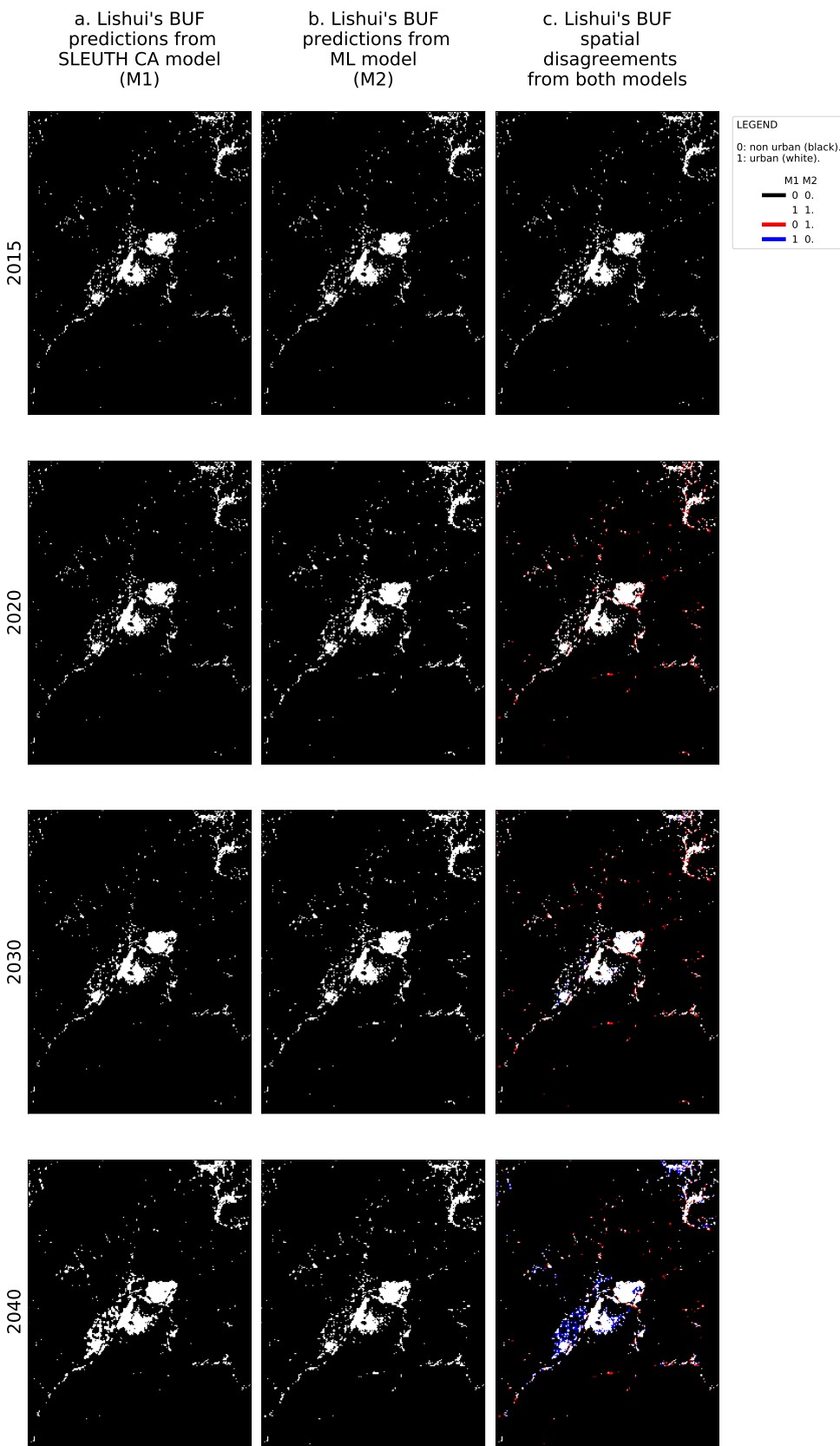

**Figure 5.** Yearly binary urban footprint predictions and spatial disagreements between the SLEUTH CA and ML models for Lishui.

Figure 6 shows for Jiaxing, the cumulative effect of the areas that changed from non-urban to urban in 2040 relative to 2015, as predicted by the SLEUTH CA model in the first column and by the ML framework in the second column. A black color in the first and second columns of these figures, represents an area that didn't change its category over time, i.e.,: that remained either as urban or as non-urban. The third column shows the spatial agreements between both models in the transition from the non-urban to urban in yellow, and their disagreements in magenta (i.e.,: when only one of the models predicted the non-urban to urban transition). Again, a black color in the third column represents areas that did not change their state over time in any of the two models. Figure 7 presents the same set of results for Lishui. Notice that in both of these figures, we used the cumulative transitions from non-urban to urban over time instead of just subtracting the urbanization of 2015 from 2040, to prevent artifacts due to de-urbanization processes.

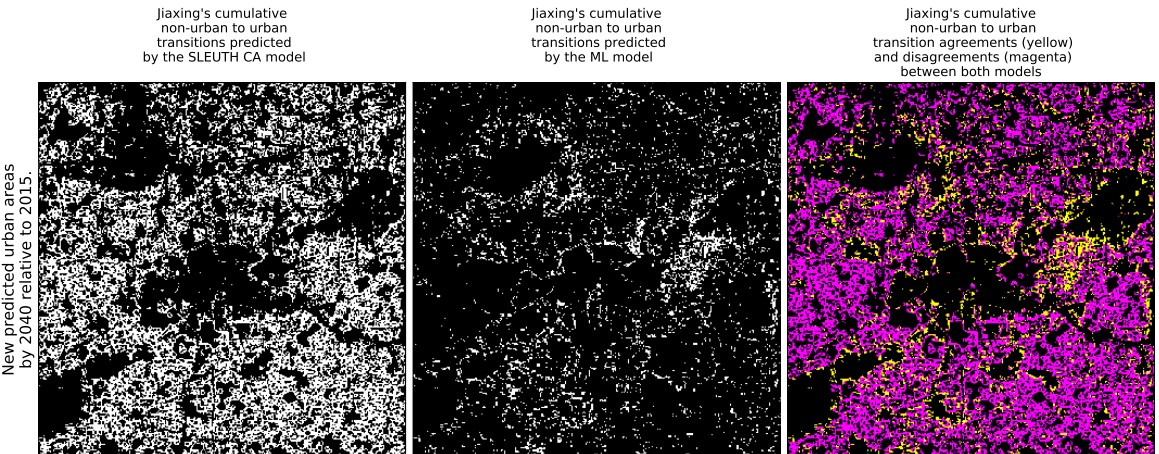

**Figure 6.** New urban areas in 2040 relative to 2015 predicted by the SLEUTH CA and ML models for Jiaxing.

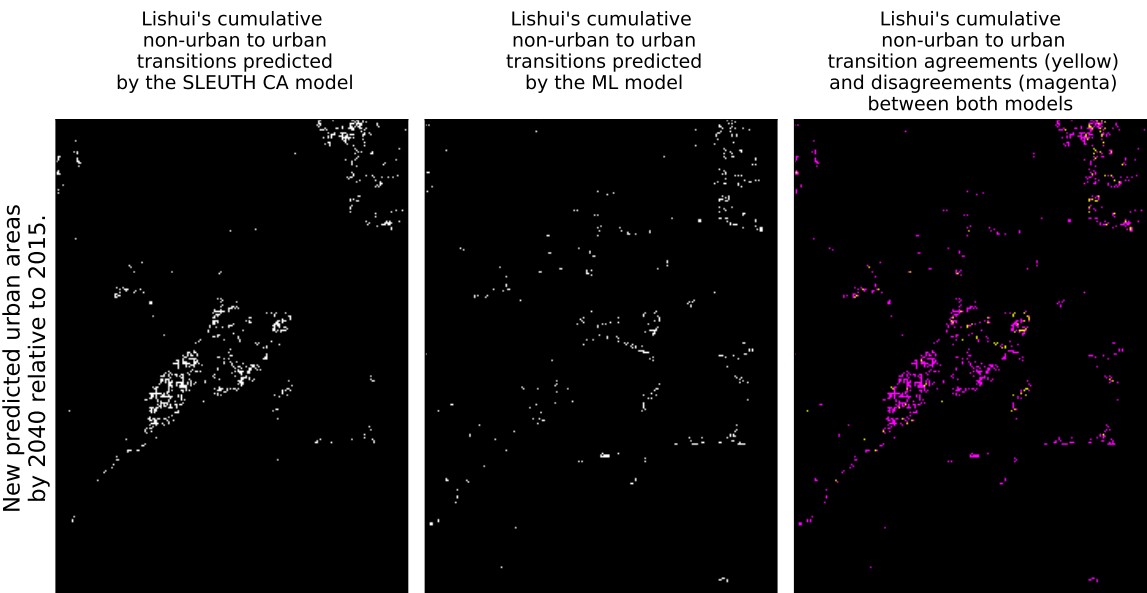

**Figure 7.** New urban areas in 2040 relative to 2015 predicted by the SLEUTH CA and ML models for Lishui.

Figures 8 and 9 present the histogram of the time difference (with sign) in the year in which the SLEUTH CA model and the ML framework predict the urbanization for Jiaxing and Lishui.

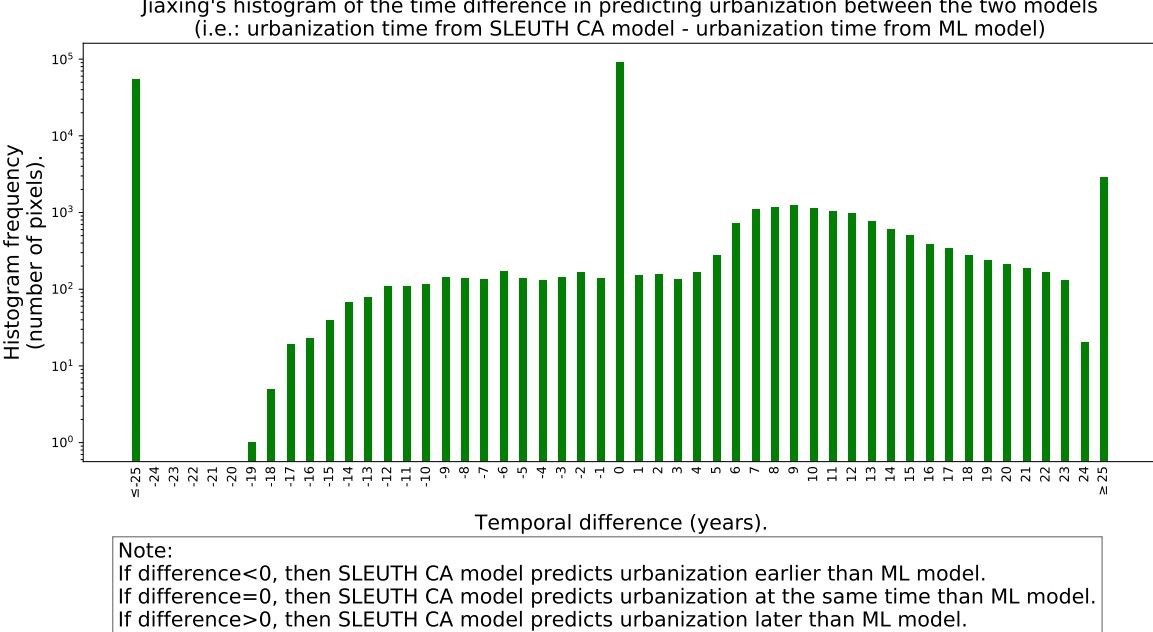

**Figure 8.** Histogram of the time difference in predicting urbanization between the two models for Jiaxing.

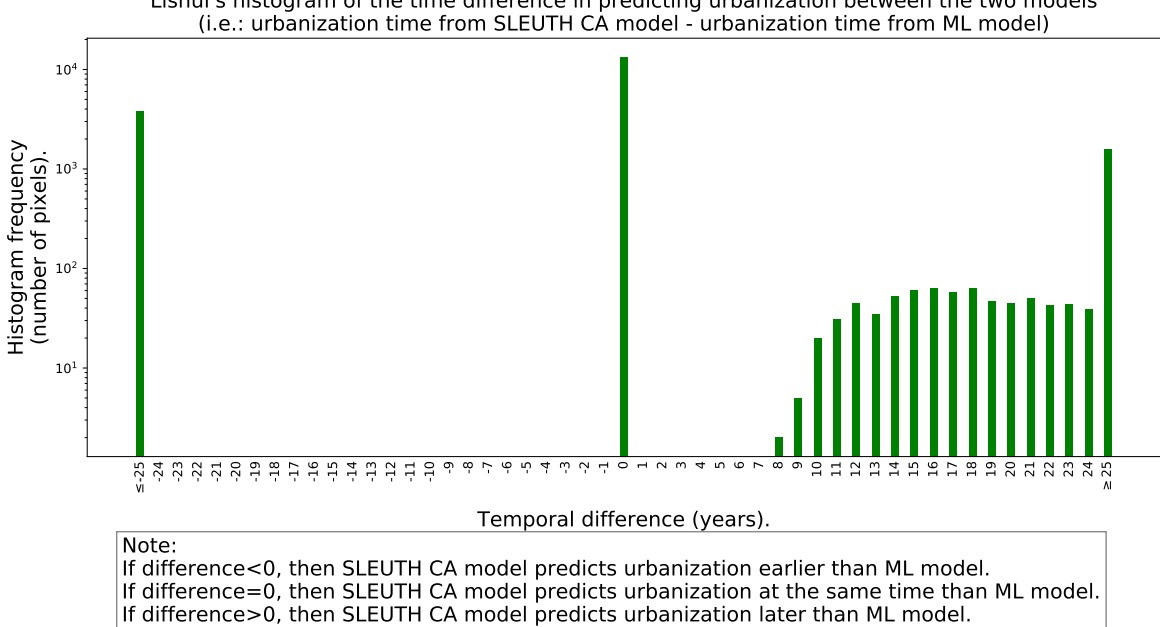

**Figure 9.** Histogram of the time difference in predicting urbanization between the two models for Lishui.

Figures 10 and 11 present the histogram of the absolute value of the time difference in the year in which the SLEUTH CA model and the ML framework predict the urbanization for Jiaxing and Lishui, respectively. Therefore, in these figures, the sign of the difference disappears.

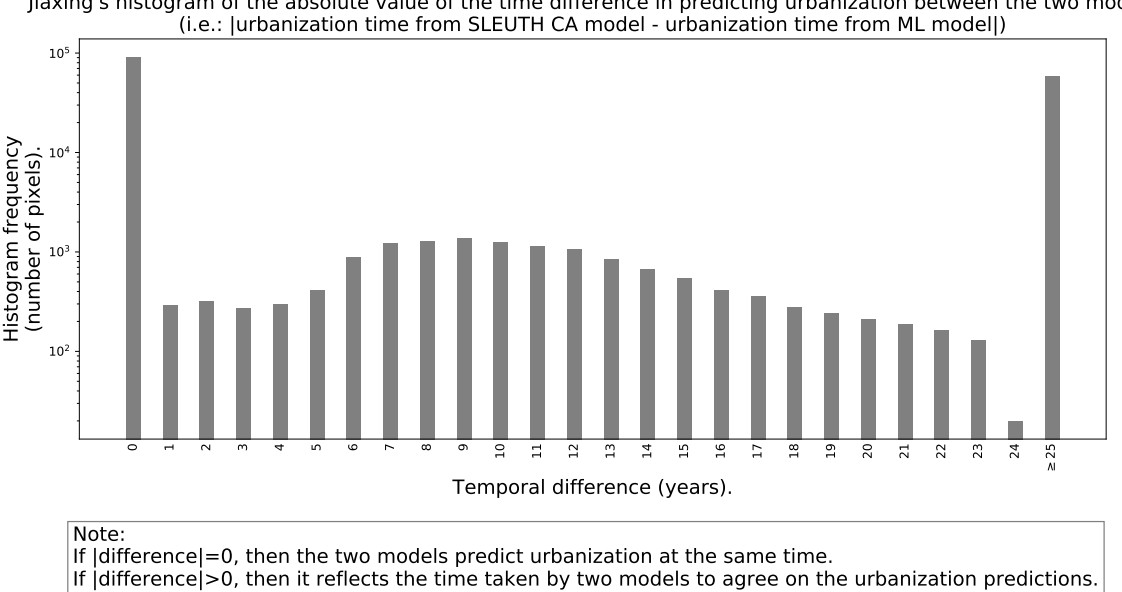

**Figure 10.** Histogram of the absolute value of the time difference in predicting urbanization between the two models for Jiaxing.

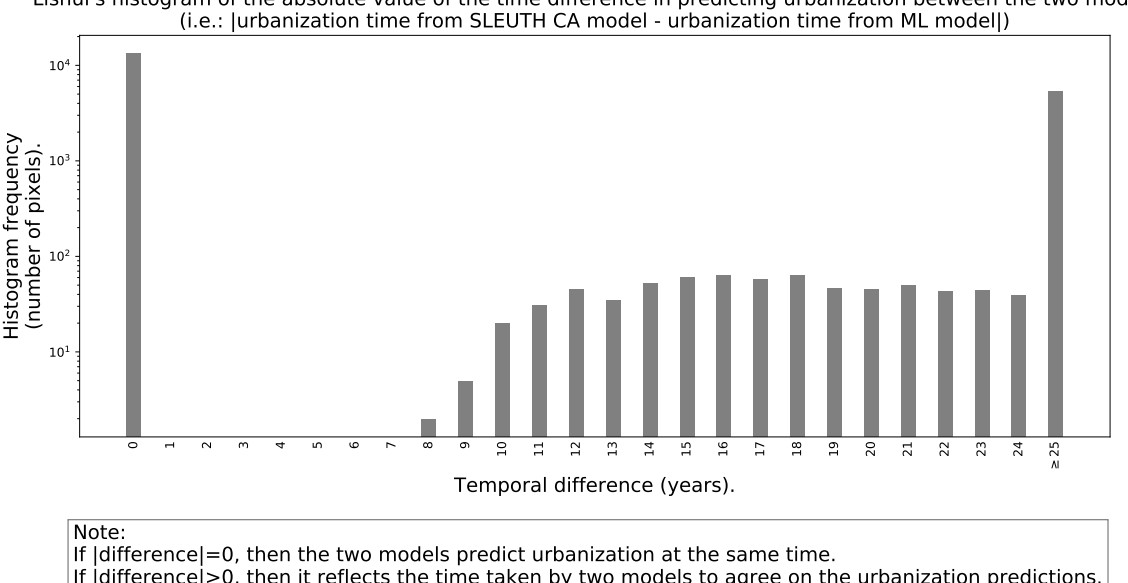

**Figure 11.** Histogram of the absolute value of the time difference in predicting urbanization between the two models for Lishui.

Figures 12 and 13 show the urban agreement ratio in the cumulative urbanization relative to 2015 over time from both models for Jiaxing and Lishui, respectively.

Figures 14 and 15 show the time series of ZNCC and SAD indices to asses the the spatial similarity between the urbanization predictions of both models for Jiaxing and Lishui, respectively.

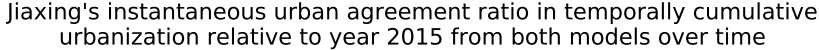

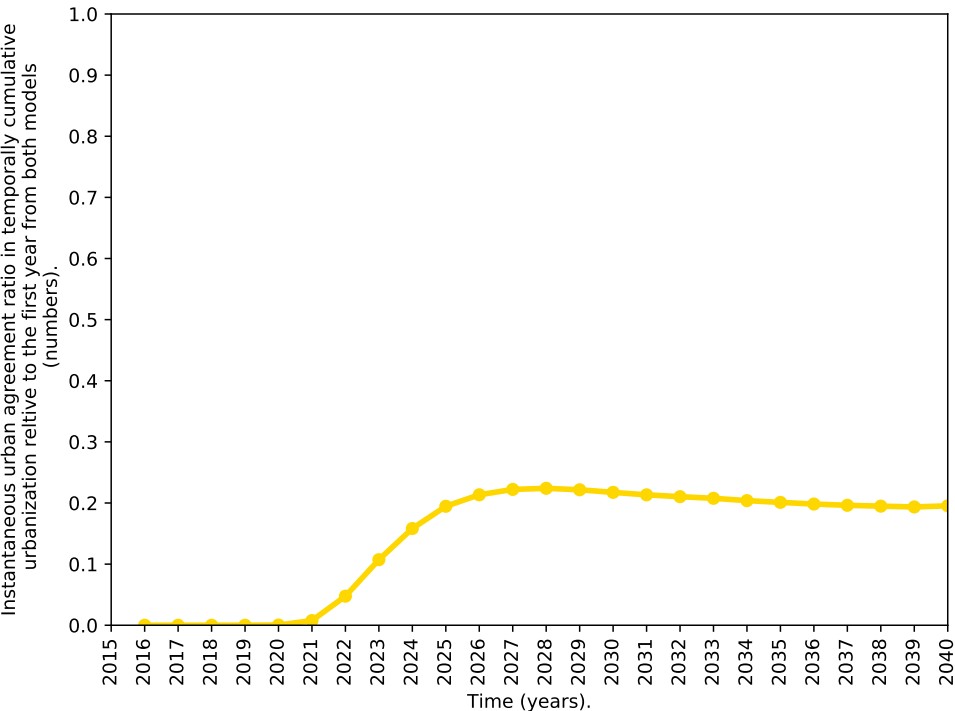

**Figure 12.** Instantaneous urban agreement ratio in temporally cumulative urbanization relative to 2015 from both models over time for Jiaxing.

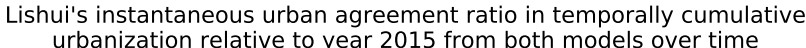

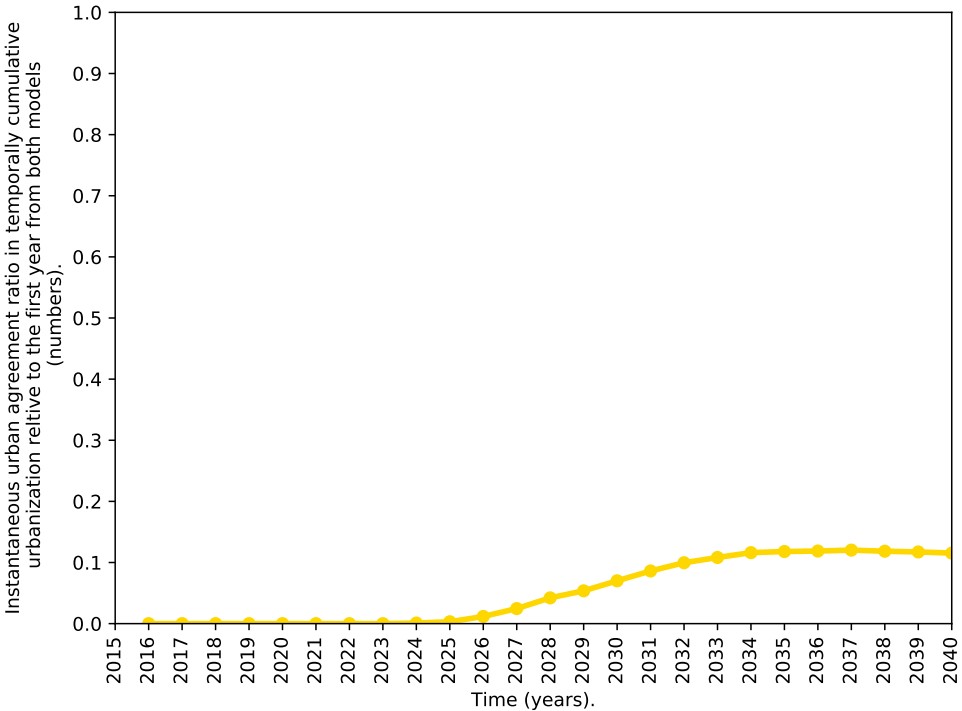

**Figure 13.** Instantaneous urban agreement ratio in temporally cumulative urbanization relative to 2015 from both models over time for Lishui.

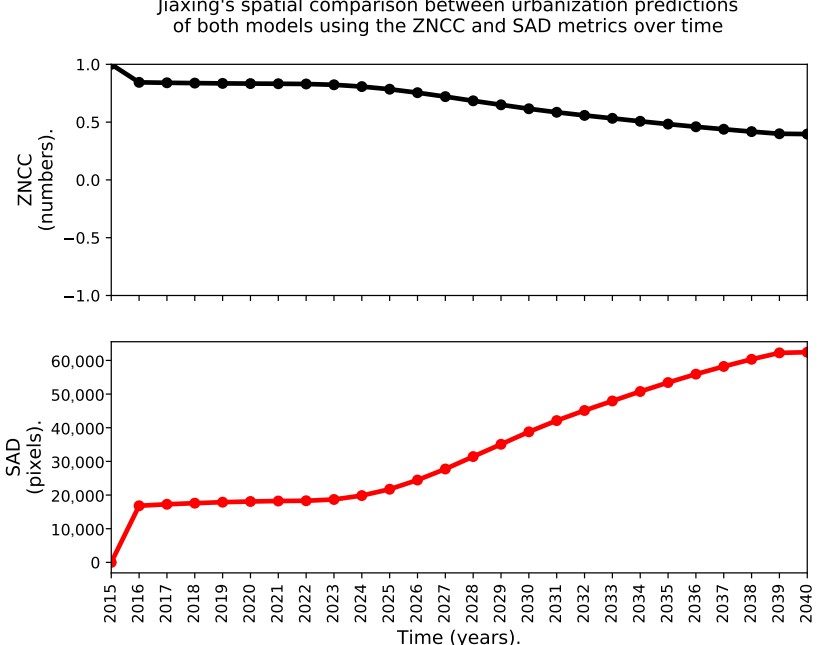

**Figure 14.** Spatial comparison between urbanization predictions of both models using the ZNCC and SAD metrics over time for Jiaxing.

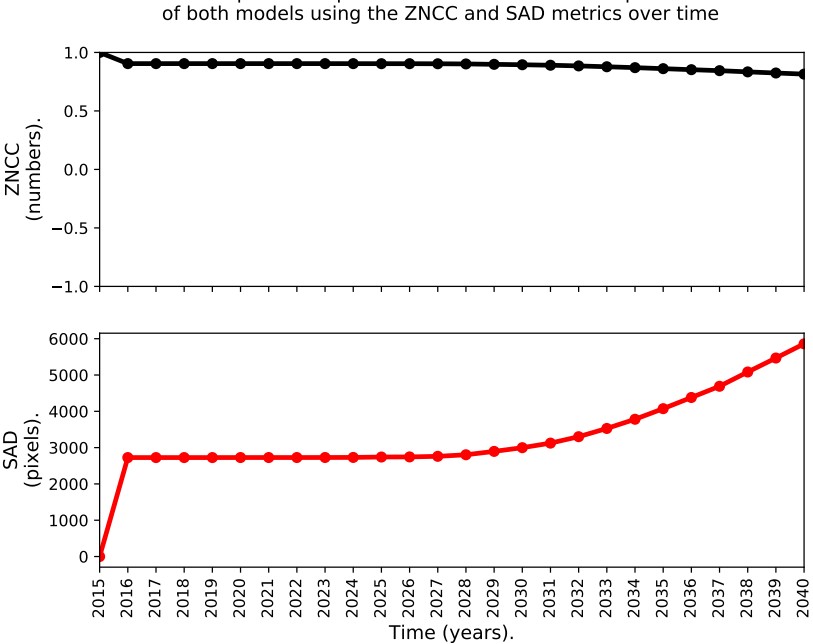

**Figure 15.** Spatial comparison between urbanization predictions of both models using the ZNCC and SAD metrics over time for Lishui.

## 4. Discussion

Calibrating the SLEUTH CA model required a certain level of human intervention, and intensive computational power. Its execution took over 2 days 10 h 16 min for Jiaxing, and 4 days 9 h 6 min for Lishui, while the predictions required 2 min for each city. The significantly larger amount of time required to train the model for Lishui can be attributed to slower convergence rate and the larger size of the input layers. Even though the coefficients range converged slowly, we had to reduce the step size for the coefficients in succeeding

iterations. Doing so substantially increased the number of runs and hence the training time. In contrast, the ML urban growth framework was highly automated and took about 7 h in a standard laptop. The exact computing time in the ML framework depends on the size of the geographic extent that contains the city to analyze, the number of input variables, and more importantly, on the number of spatial windows, temporal lags, machine learning models, and hyper-parameters to explore.

Visual inspection of Figures 2 and 3, as well as Figures 4 and 5, reveals that there are significant variations between the simulation results of the two models. In the long term, the SLEUTH CA model predicts larger urban areas than the ML framework in both cities. However, the ML framework predicts most of the new urbanization in the next decade or so, whereas the SLEUTH CA model predicts most of its growth till the last decades of analysis. The urbanization predictions by each model in both cities are slightly different. In Jiaxing, the SLEUTH CA model predicts a more compact growth, while the ML framework predicts a more scattered growth that manages to maintain the city structure. In Lishui, both urban growth predictions are fairly fragmented, but overall, the ML model predicts most of such growth in the periphery whereas the SLEUTH CA predicts that it will happen near the urban center.

To simplify the uncertainty analysis, we define "certain areas in space" as those regions where both models predicted the same results (urban or non-urban) but not necessarily at the exact same time. This is represented in yellow in Figures 6 and 7 for both cities in the year 2040. Similarly, we define "certain areas in time" as a subset of the previous cases when both models predict the same results at the same time. By looking at Figures 12 and 13, we find that the agreement between the two models is far better in Jiaxing than in Lishui, and yet, it never surpasses thirty percent. We should highlight that certain areas in time can help to prioritize areas by policy makers to layout infrastructure that need long lead times, which are often required to better position small cities. Moreover, the certain areas in space and time can also be applied to promote the development of retail and commercial clusters, provision of kindergartens and elementary schools, and other urban amenities in order to avoid bedroom community and ghost town type of urban growth in China.

By analyzing the histograms of the signed time difference in the prediction of urbanization between the two models displayed in Figures 8 and 9, we see that the heights of the histogram for negative differences are mostly lower than the heights of the histogram for the corresponding positive differences. However, the histogram shape in the case of Jiaxing is not as asymmetric with respect to zero as it is in Lishui. A positive temporal difference in the signed histogram, means that the SLEUTH CA model predicts urbanization later than the ML framework. In both cities, the histograms of signed differences show at least three well-defined peaks. The peak at zero indicates how many pixels are predicted to be urban at the same time for both models. The peak at $\leq -25$ reflects that the SLEUTH CA model predicts as urban land some areas that are never predicted as such by the ML framework in the simulation horizon. Conversely, the peak at $\geq 25$ years represents some areas that are predicted as urban land by the ML framework that are never predicted as urban by the SLEUTH CA in the simulation horizon. If the prediction models were very similar, the histograms of signed differences would resemble a normal distribution centered around zero with a very small standard deviation. Looking at Figures 10 and 11, we see that in Jiaxing, the models reach some agreement about urbanization predictions progressively but it takes a while as the histogram is spread between zero and twenty four years. However, in Lishui the agreement pattern has a strong transition, which could be understood as a temporal lag between the models' predictions that only starts to improve after eight years. The instantaneous urban agreement ratio in temporally cumulative urbanization relative to 2015 depicted in Figures 12 and 13 shows a slow improvement over time in both cities. This behavior was somewhat expected because of the fixed geographic extent used for the predictions. The explanation is as follows, during the earlier years there is plenty of space to urbanize, but as time advances, the available land becomes scarcer, and one of the urban growth models will predict as urban land some areas that had been predicted in the past as

urban by the second model. Interestingly, in Jiaxing, this metric grows much faster than in Lishui. The main reason for the differences in these two cities is that in Jiaxing both urban growth models predict lots of new urban areas, but in Lishui there is an unbalance, as the SLEUTH CA predicts much more urbanization than the ML framework.

Figures 14 and 15 show that both models have a better agreement in Lishui than in Jiaxing. At first glance, this conclusion seems to contradict our previous analysis, but it does not. It just reflects the fact that both the ZNCC and the SAD metrics are too influenced by the existing urban areas that were already in place for the last historical year in the data, 2015 in our case. Both metrics perform better (i.e.,: a ZNCC closer to one and a SAD value closer to zero) in larger areas of analysis experiencing little urban growth over time like Lishui than in smaller areas of analysis with fast-paced urbanization like Jiaxing. With these considerations in mind, we think that the agreement index $S$, modeled through Equations (2) and (3), is a stronger metric than the ZNCC for assessing the similarity of the predictions between two urban growth models.

Unlike the SLEUTH CA model that only focuses on the existing and the past conditions of urbanization, the ML framework also considers and predicts the spatially distributed population growth, and therefore, if the predicted population can be allocated within the existing urban fabric, the ML framework does not promote new urban areas. Similarly, if the population growth inside a non-urban area is slow, it can take a while before it marks the land as urban. After analyzing both cities, the current predictions for Jiaxing might help policymakers to prioritize their agendas for urban interventions, however, in the case of Lishui, it is clear that further studies are required to reduce the uncertainty before committing to long-term infrastructure plans.

Under the TDSC policy, small cities are gaining the upper hand for rapid growth. Forecasting the uncertainty of their growth patterns in space and time are critical for policy makings. For Jiaxing, the uncertainty lies in the type of urban growth. For example, a more sporadic pattern can induce sprawl, while a more concentrated and compact pattern can create land use right and equality issues. For Lishui, the uncertainty is hinged upon the rate of growth. The essence of the growth speed is determined by the balance of economic growth competition and environmental preservation. As the TDSC policies are going beyond the boundary of Zhejiang Province, hundreds of small cities in China will need to consider the spatiotemporal urban growth uncertainty in their long-term planning. Consequently, the space and time uncertainty modeling framework could provide empirical evidence for the transformation of urban networks in China. Beyond the Chinese context, the uncertainty framework can also be tailored to reflect context-specific urban growth conditions around the globe.

## 5. Conclusions

In this paper, we analyzed the spatiotemporal uncertainty in urban growth through a comparison of the agreements and disagreements in the predictions of two models, a SLEUTH CA that works with rules and encodes human intuition, and a machine learning framework that is purely data-driven. The proposed methodology was tested in the Chinese cities of Jiaxing and Lishui as case studies. We chose these two cities because they have different conditions and are part of the group of smaller cities in China that are expected to grow in the upcoming decades under the new TDSC policies. We focused on the spatiotemporal uncertainty, understood as the inherent doubt in the predictions of where and when will a piece of land become urban in the territory. This is important for urban planning policy making because it can help to localize and prioritize investments for public infrastructure and facilities. The resulting uncertainty measures can shed some light on the required effort and urgency of urban interventions, as they depend on the spatial extent and available time to complete the works. The main benefit of our analyses is that it can be applied worldwide at low cost, as most of the required variables are in suitable digital formats, and consolidated in public repositories for free, except commonly for the land-use land-cover change data. The proposed methodology for understanding the

spatiotemporal uncertainty can use other pairs of growth models. However, we argue that in any scenario, it is desirable to select urban growth models from different families and not just different instances of the same family of models to provide a stronger assessment. So far, one of the limitations of the prediction models that we used is that they are ill-equipped to anticipate the effects of rare or black swan events (i.e.,: a sudden change in the urbanization rate due to pandemics, strong migration waves, change of administrative boundaries, economic recession, etc.). There are three main reasons. Firstly, if a similar event never occurred or was not included in the input dataset, the models have no way of knowing which features to look at for their prediction [4]. Secondly, the associated effects of rare events can manifest themselves at different temporal delays in the variables of interest [36], which makes assembling a representative dataset even harder. Thirdly, the explored predictive models are based on simplified ideal forms, which ignore the incredible complexity of reality. However, and despite the models' limitations in forecasting rare events, their estimations may still have some constant properties useful for extrapolation and forecast [37]. Therefore, we recommend either re-calibrating or re-training the models with new data once a year or so and then update the urban estimates, independent of the urban growth model pair to use. In this way, interested users can extend the validity of urban predictions and their uncertainties. As future work, we want to investigate an extension of the proposed spatiotemporal analysis within regional urban networks. We also want to fuse the predictions of different urban growth models, considering their relative strengths and weaknesses, for improving the overall urbanization forecast accuracy.

**Author Contributions:** Conceptualization, J.A.G. and C.G.; methodology, C.G. and J.A.G.; software, J.A.G., C.G., P.T., S.P. and J.L.; validation, J.A.G., C.G., P.T. and S.P.; formal analysis, J.A.G. and C.G.; investigation, J.A.G., C.G., P.T. and J.L.; resources, M.K., J.C.D.; data curation, S.P., P.T. and J.L.; writing—original draft preparation, J.A.G. and C.G.; writing—review and editing, J.C.D. and M.K.; visualization, J.A.G., C.G. and S.P.; supervision, J.C.D.; project administration, J.C.D. and M.K.; funding acquisition, M.K. All authors have read and agreed to the published version of the manuscript.

**Funding:** This document was completed with support from the PEAK Urban programme, supported by UKRI's Global Challenge Research Fund, Grant Ref: ES/P011055/1.

**Institutional Review Board Statement:** Not applicable.

**Informed Consent Statement:** Not applicable.

**Data Availability Statement:** Publicly available datasets were analyzed in this study. This data can be found here: https://github.com/Rise-group/urban_growth_uncertainty_China_with_SLEUTH_CA_and_ML_models.

**Acknowledgments:** Authors want to thank the support from the PEAK Urban Programme, EAFIT University, Universidad Icesi, NYU Shanghai, Indian Institute for Human Settlements, and University of Oxford.

**Conflicts of Interest:** The authors declare no conflict of interest.

## Abbreviations

The following abbreviations are used in this manuscript:

| | |
|---|---|
| GHSL | Global human settlement layer |
| BUF | Binary urban footprint |
| POP | Population distribution |
| LULCC | Land-uses and land-cover changes |
| DEM | Digital elevation map |
| SAD | Sum of absolute differences |

| | |
|---|---|
| SSD | Sum of square differences |
| MSE | Mean squared error |
| RMSE | Root mean square error |
| FP | False positive |
| ZNCC | Zero-mean normalized cross correlation |
| IoU | Intersection over union |
| KDE | Kernel density estimation |
| PDF | Probability density function |
| CA | Cellular automaton |
| ML | Machine learning |

## Appendix A. Auxiliary Tables for SLEUTH CA Model Calibration

**Table A1.** Coefficient values [start, stop, step] of parameters at each step of the SLEUTH CA model.

| Stage | Monte Carlo Iteration | Diffusion | Breed | Spread | Slope | Road Gravity |
|---|---|---|---|---|---|---|
| Lishui: size of each image layer 704 rows × 518 columns. | | | | | | |
| Coarse | 5 | [0, 100, 25] | [0, 100, 25] | [0, 100, 25] | [0, 100, 25] | [0, 100, 25] |
| Fine | 8 | [25, 100, 15] | [1, 100, 10] | [13, 38, 5] | [75, 100, 5] | [50, 100, 10] |
| Final | 10 | [25, 100, 5] | [41, 61, 2] | [11, 15, 1] | [75, 100, 2] | [50, 100, 5] |
| Derive coeff. | 150 | [25, 25, 1] | [53, 53, 1] | [13, 13, 1] | [77, 77, 1] | [95, 95, 1] |
| Predict | 200 | 32 | 67 | 17 | 68 | 96 |
| Jiaxing: size of each image layer 451 rows × 442 columns. | | | | | | |
| Coarse | 5 | [0, 100, 25] | [0, 100, 25] | [0, 100, 25] | [0, 100, 25] | [0, 100, 25] |
| Fine | 8 | [88, 100, 4] | [13, 38, 5] | [13, 38, 5] | [25, 100, 15] | [25, 100, 15] |
| Final | 10 | [92, 100, 1] | [13, 38, 2] | [18, 28, 2] | [85, 100, 3] | [25, 100, 5] |
| Derive coeff. | 150 | [98, 98, 1] | [15, 15, 1] | [26, 26, 1] | [85, 85, 1] | [65, 65, 1] |
| Predict | 200 | 100 | 15 | 27 | 1 | 74 |

**Table A2.** Top three rows of the 'control_stats.log' file sorted by the OSM metric. Diff = Diffusion; Brd = Breed; Sprd = Spread; Slp = Slope; RG = Road Gravity.

| Lishui | | | | | | Jiaxing | | | | | |
|---|---|---|---|---|---|---|---|---|---|---|---|
| OSM | Diff | Brd | Sprd | Slp | RG | OSM | Diff | Brd | Sprd | Slp | RG |
| Coarse calibration, runs: 3125, time: 1 h 18 m. | | | | | | Coarse calibration, runs: 3125, time: 1 h 31 m. | | | | | |
| 0.593 | 25 | 1 | 25 | 75 | 50 | 0.71 | 100 | 25 | 25 | 100 | 25 |
| 0.581 | 25 | 100 | 25 | 100 | 100 | 0.711 | 100 | 25 | 25 | 100 | 50 |
| 0.577 | 100 | 50 | 25 | 100 | 75 | 0.639 | 100 | 25 | 25 | 25 | 100 |
| Fine calibration, runs: 12,960, time: 7 h 37 m. | | | | | | Fine calibration, runs: 5184, time: 3 h 22 m. | | | | | |
| 0.751 | 25 | 41 | 13 | 75 | 100 | 0.844 | 100 | 13 | 28 | 85 | 25 |
| 0.748 | 25 | 61 | 13 | 80 | 90 | 0.829 | 100 | 13 | 28 | 100 | 70 |
| 0.741 | 100 | 51 | 13 | 100 | 50 | 0.824 | 92 | 38 | 18 | 85 | 100 |
| Final calibration, runs: 125,840, time: 4 d 11 m. | | | | | | Final calibration, runs: 67,392, time: 2 d 9 h 52 m. | | | | | |
| 0.764 | 25 | 53 | 13 | 77 | 95 | 0.841 | 98 | 15 | 26 | 85 | 65 |
| 0.760 | 40 | 51 | 13 | 85 | 60 | 0.838 | 100 | 13 | 28 | 85 | 25 |
| 0.755 | 25 | 55 | 14 | 79 | 70 | 0.834 | 100 | 13 | 28 | 88 | 35 |

## Appendix B. Additional Figures for the Ml-Based Urban Growth Framework

We used the historical records between 1990 and 2015 to understand the growth dynamics in Jiaxing and Lishui through three metrics, the total urban population, the total

urban area, and the urban density. We calculated them by counting the population inside the binary urban footprint, summing the number of urban pixels in the binary urban footprint and multiplying the result by the pixel size, and computing the ratio between the two former values. The time series with these metrics are shown in Figures A1 and A2 for Jiaxing and Lishui, respectively.

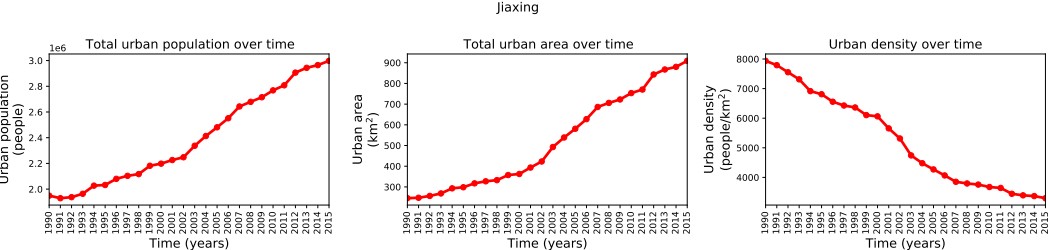

**Figure A1.** Diagnostic graphs of the geographic area of Jiaxing. (**Left**) total urban population. (**Center**) total urban area. (**Right**) urban density.

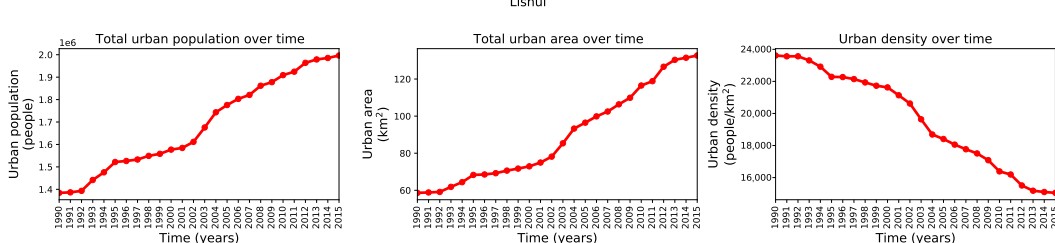

**Figure A2.** Diagnostic graphs of the geographic area of Lishui. (**Left**) total urban population. (**Center**) total urban area. (**Right**) urban density.

We estimated the probability density function of a population threshold after which a non-urban pixel becomes urban for Jiaxing and Lishui. We include the results in Figures A3 and A4.

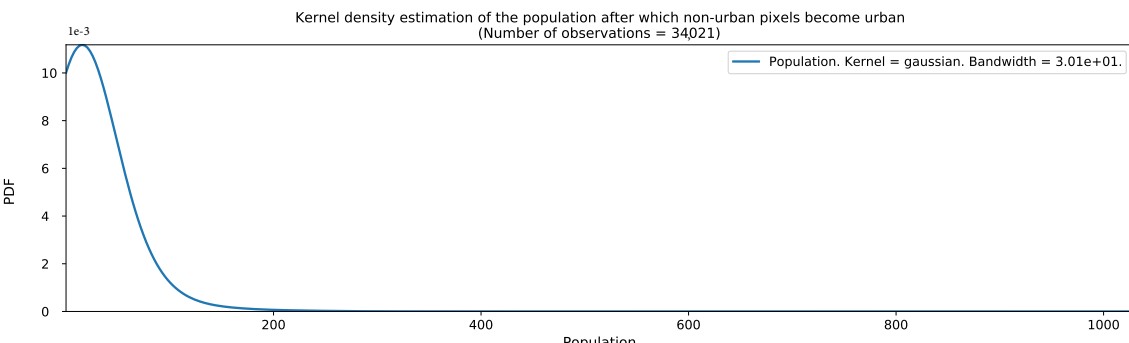

**Figure A3.** Estimated probability density function of a population threshold after which a non-urban pixel becomes urban in Jiaxing.

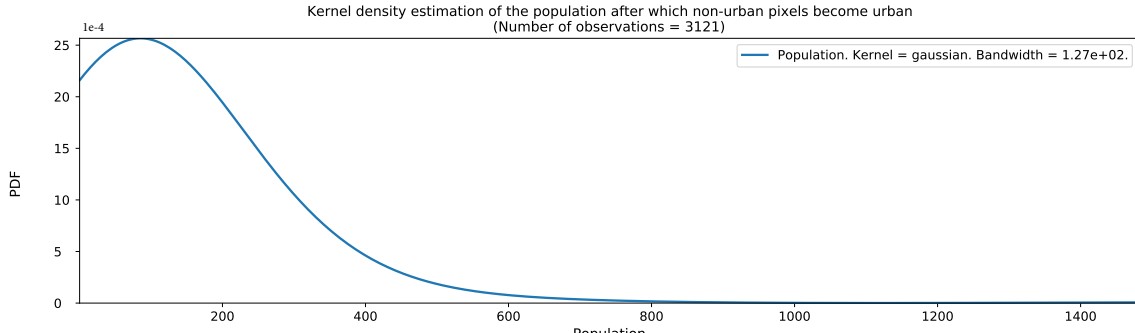

**Figure A4.** Estimated probability density function of a population threshold after which a non-urban pixel becomes urban in Lishui.

Following the procedure described in [23], we computed population threshold maps that we used for determining when a non-urban pixel becomes urban for Jiaxing and Lishui. We present the results in Figures A5 and A6.

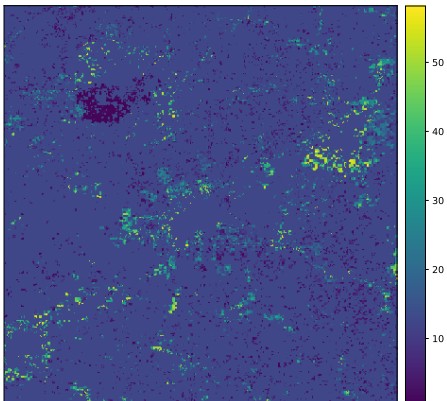

**Figure A5.** Selected population threshold after which a non-urban pixel becomes urban for Jiaxing.

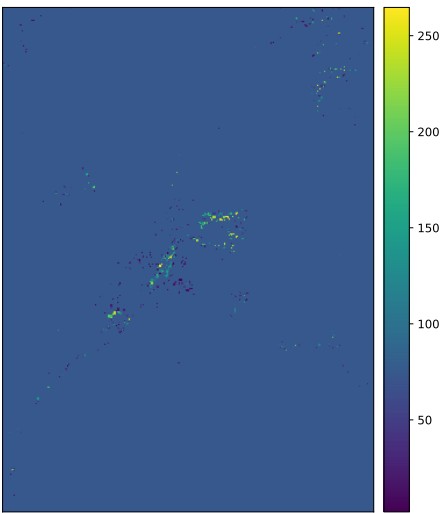

**Figure A6.** Selected population threshold after which a non-urban pixel becomes urban for Lishui.

For the years in the test set (2011–2015), we computed the urban growth framework performance in terms of the errors in population distribution and binary urban footprint estimations relative to the ground truth values for Jiaxing and Lishui. Figure A7 shows the corresponding results. Notice that the data from 2011 and 2012 were used to predict 2013, and similarly, the data from 2012 and 2013 were used to predict 2014, and the data from 2013 and 2014 were used to predict 2015.

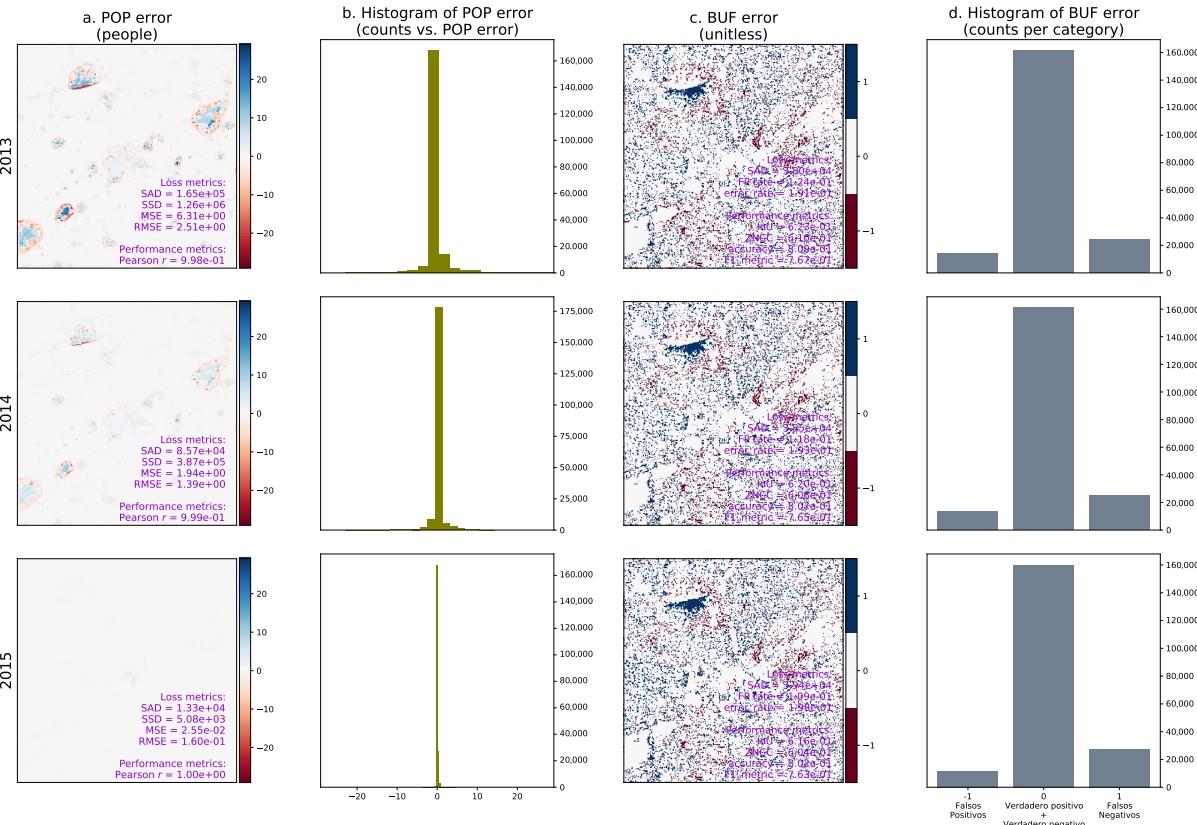

(**a**) Jiaxing.

**Figure A7.** *Cont.*

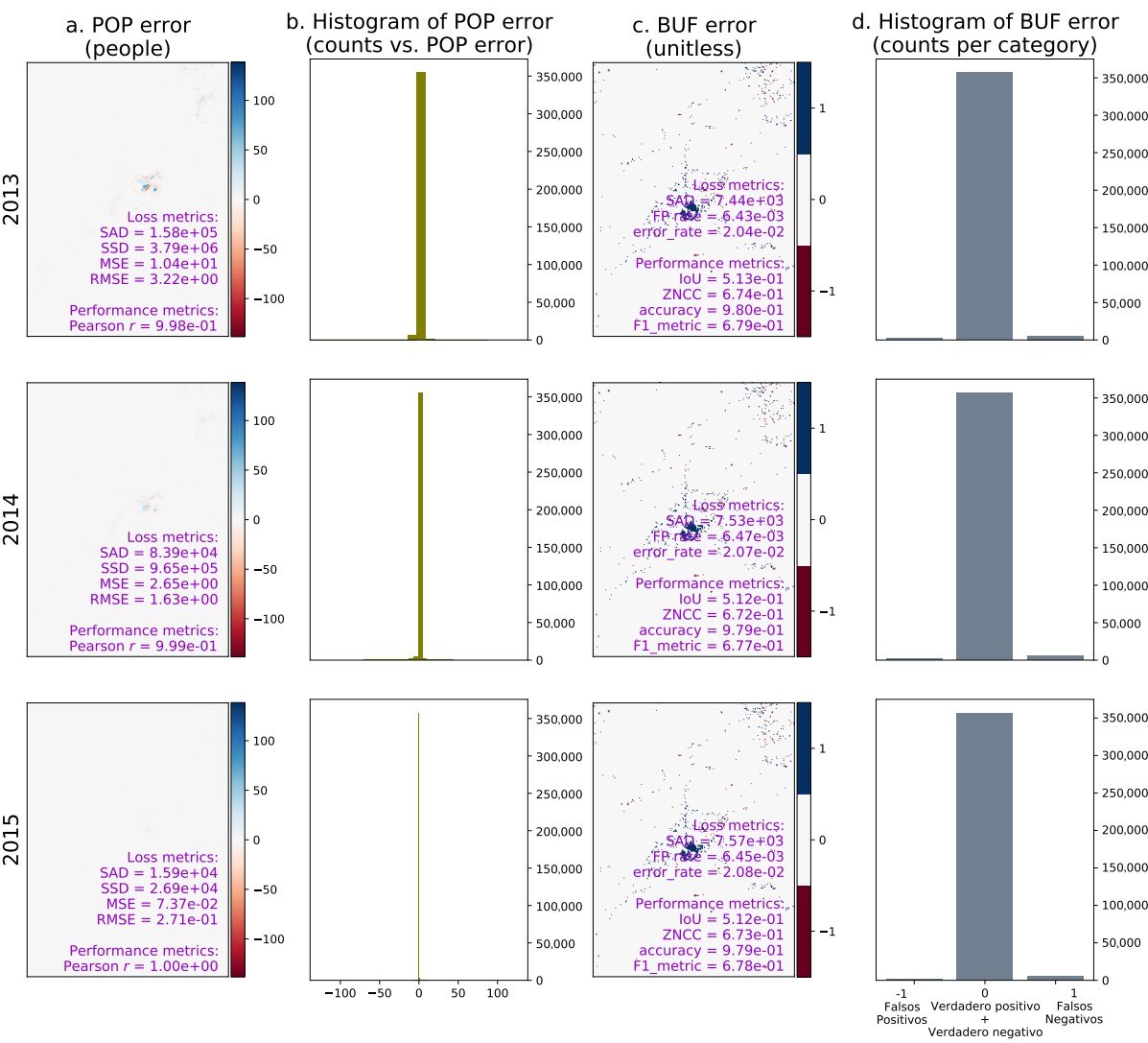

(**b**) Lishui.

**Figure A7.** Framework performance in the geographic areas of interest. From left to right, the columns correspond to: (1) population distribution error (i.e., real value minus predicted value); (2) histogram of the population distribution error; (3) binary urban footprint error (i.e., real value minus predicted value); (4) histogram of the binary urban footprint error. POP = predicted population distributions; BUF = binary urban footprints; ZNCC = zero-mean normalized cross-correlation; SAD = sum of absolute differences; SSD = sum of squared differences.

## Appendix C. Land Use Recoding for Jiaxing and Lishui

To use the SLEUTH CA model, the original land-use land-cover change (LULCC) information from Jiaxing and Lishui was recoded using the Anderson Level I classification system. The applied conversion is summarized in Table A3, where the new categories for the LULCC after this recoding are given by:

1. Urban.
2. Agricultural.
3. Rangeland.
4. Forest Land.
5. Water.
6. Wetland.
7. Barren Land.
8. Tundra. This category was not available in Jiaxing nor Lishui.

9. Perennial Snow or Ice. This category was not available in Jiaxing nor Lishui.

**Table A3.** Land use recoding for Jiaxing and Lishui.

| Old Parent Class | Old Class | Pixel Value | New Class | Pixel Value |
|---|---|---|---|---|
| Grassland | Grass | 23 | Rangeland | 3 |
| | Herbaceous green space | 24 | Rangeland | 3 |
| Wetlands | Herbaceous wetlands | 33 | Wetland | 6 |
| | Lake | 34 | Water | 5 |
| | Reservoir/pit | 35 | Water | 5 |
| | River | 36 | Water | 5 |
| | Canal | 37 | Water | 5 |
| Arable land | Paddy field | 41 | Agriculture | 2 |
| | Dry land | 42 | Barren land | 7 |
| Artificial Surface | Residential | 51 | Urban | 1 |
| | Industrial | 52 | Urban | 1 |
| | Transportation | 53 | Urban | 1 |
| | Mining farm | 54 | Urban | 1 |
| Other | Bare rock | 65 | Barren land | 7 |
| | Bare soil | 66 | Rangeland | 3 |
| Woodland | Evergreen broad-leaved forest | 101 | Forest land | 4 |
| | Deciduous broad-leaved forest | 102 | Forest land | 4 |
| | Evergreen coniferous forest | 103 | Forest land | 4 |
| | Deciduous coniferous forest | 104 | Forest land | 4 |
| | Coniferous and broad-leaved mixed forest | 105 | Forest land | 4 |
| | Evergreen broad-leaved shrub forest | 106 | Forest land | 4 |
| | Deciduous broad-leaved forest | 107 | Forest land | 4 |
| | Evergreen coniferous forest | 108 | Forest land | 4 |
| | Arbor | 109 | Rangeland | 3 |
| | Bush field | 110 | Rangeland | 3 |
| | Arbor green space | 111 | Rangeland | 3 |
| | Shrubland | 112 | Rangeland | 3 |

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
