# Peer review of "Analyzing the Spatiotemporal Uncertainty in Urbanization Predictions"

_remotesensing, doi:10.3390/rs13030512_

Round 1

Reviewer 1 Report

It is an interesting paper, well written, but the conclusions needs to be implemented as follows:

The authors say: "one of the biggest limitations of most prediction models, is that they are ill-equipped to anticipate the effects of rare or black swan events...". They recommend to re-calibrating or re-training theese models and we would like to understand also at this level how they think to adapt the models and above all if the models are able to explain the changes that occur in time and space. It is also a philosophical issue that cannot be explained only with models but of which we need to be aware.

Author Response

Dear reviewer, please find our response in the attached document.

Reviewer 2 Report

The article approaches an up-to-date problem and focuses the topic on a very interesting point of view.

We suggest, and guess, a further in-depth after pandemic to better understand in what way the black swan event could have modified the prevision. 

Author Response

(The authors gave the same response as above.)

Reviewer 3 Report

In general, this paper has seemed like a good work, both in the research approach and in the development and the conclusions provided.

It is an interesting research that can contribute to studies of evolution and growth of urban developments.

I would like to clarify a few aspects:

   1.- In the Introduction sections, you talk about the cities to be analyzed and repeat data that are then presented again in the next section.

   2.- In Figure 1, both images do not provide much information, it could be completed with graphic information such as the delimitation of the cities currently, the neighboring cities,...You could also add some more toponyms. Even the representation of the relief, throught  some form of expression (contour lines, etc.) would provide more information.

    3.- In Table 2, all the variables used in the models are compiled, however, the spatial resolution of each variable used should be justified, and therefore the spatial accuracy, as for example occurs with Terrain slope, Hillshade, Water bodies. This justification should be linked to its influence on the results of the models.

    4.- In section 2.1 you establish that you are going to work with raster information in a size of 100 * 100, when there are variables that intervene in the models with spatial resolutions of 250 * 250 m. This should be better justified.

Author Response

(The authors gave the same response as above.)
